# Structure and Luminescent Properties of Niobium-Modified ZnO-B_2_O_3_:Eu^3+^ Glass

**DOI:** 10.3390/ma17061415

**Published:** 2024-03-20

**Authors:** Reni Iordanova, Margarita Milanova, Aneliya Yordanova, Lyubomir Aleksandrov, Nikolay Nedyalkov, Rositca Kukeva, Petia Petrova

**Affiliations:** 1Institute of General and Inorganic Chemistry, Bulgarian Academy of Sciences, G. Bonchev, Str., Bld. 11, 1113 Sofia, Bulgaria; reni@svr.igic.bas.bg (R.I.); a.yordanova@svr.igic.bas.bg (A.Y.); lubomir@svr.igic.bas.bg (L.A.); rositsakukeva@yahoo.com (R.K.); 2Institute of Electronics, Bulgarian Academy of Sciences, Tzarigradsko Shousse 72, 1784 Sofia, Bulgaria; nned@ie.bas.bg; 3Institute of Optical Materials and Technologies “Acad. Jordan Malinowski”, Bulgarian Academy of Sciences, Blvd. Akad. G. Bonchev Str., Bld. 109, 1113 Sofia, Bulgaria; petia@iomt.bas.bg

**Keywords:** glass structure, europium, IR, photoluminescence, density

## Abstract

The effect of the addition of Nb_2_O_5_ (up to 5 mol%) on the structure and luminescent properties of ZnO-B_2_O_3_ glass doped with 0.5 mol% (1.32 × 10^22^) Eu_2_O_3_ was investigated by applying infrared (IR), Raman and photoluminescence (PL) spectroscopy. Through differential thermal analysis and density measurements, various physical properties such as molar volume, oxygen packing density and glass transition temperature were determined. IR and Raman spectra revealed that niobium ions enter into the base zinc borate glass structure as NbO_4_ tetrahedra and NbO_6_ octahedra. A strong red emission from the ^5^D_0_ level of Eu^3+^ ions was registered under near UV (392 nm) excitation using the ^7^F_0_ → ^5^L_6_ transition of Eu^3+^. The integrated fluorescence intensity ratio R (^5^D_0_ → ^7^F_2_/^5^D_0_ → ^7^F_1_) was calculated to estimate the degree of asymmetry around the active ion, suggesting a location of Eu^3+^ in non-centrosymmetric sites. The higher Eu^3+^ luminescence emission observed in zinc borate glasses containing 1–5 mol% Nb_2_O_5_ compared to the Nb_2_O_5_-free zinc borate glass evidences that Nb_2_O_5_ is an appropriate component for modifying the host glass structure and improving the emission intensity.

## 1. Introduction

Glasses accommodating rare-earth ions have been studied for years as luminescent materials in solid-state lasers, photonics, and opto-electronic devices like optical amplifiers, multicolor displays and detectors. Among them, glasses containing trivalent europium ion have been the subject of a great deal of interest due to its intense red emission [1,2,3,4,5]. Currently, heavy emphasis has been given to the discovery of new glass compositions for exploitation as Eu^3+^-doped luminescent hosts, as the optical properties of the active rare-earth ions in glasses strongly depend on the chemical composition of the glass matrix [6]. Glasses containing Nb_2_O_5_ are suitable matrices for doping with active Eu^3+^ ions since Nb^5+^ ions can modify the environment around the rare-earth ions due to their higher polarizability [7]. Also, Nb_2_O_5_ possesses significant optical characteristics, such as low phonon energy, high refractive index (*n* = 2.4), NIR and visible transparency, that are directly related to the luminescence properties [8,9]. The optical properties and glass-forming ability of Nb_2_O_5_-containing glasses are strongly related with the structural features of glasses and more particularly with the coordination state of Nb^5+^ ions and their way of bonding in the glass network, making the structural role of Nb_2_O_5_ in various glass compositions also a subject of intensive research. IR and Raman spectroscopic studies indicate that the niobium present in the amorphous network in the form of octahedral NbO_6_ units or NbO_4_ tetrahedral groups with different degrees of distortions and types of bonding (by corners and by edges) [10,11,12].

In this work, we report on the preparation, structure and photoluminescence properties of glasses 50ZnO:(50 − x)B_2_O_3_:0.5Eu_2_O_3_:xNb_2_O_5_, (x = 0, 1, 3 and 5 mol%). The aim is to investigate the effect of the addition of Nb_2_O_5_ to the binary 50ZnO:50B_2_O_3_ glass, on the glass structure and photoluminescence properties of the active Eu^3+^ ions doped in this host glass matrix.

## 2. Materials and Methods

Glasses with the composition in mol% of 50ZnO:(50 − x)B_2_O_3_:xNb_2_O_5_:0.5Eu_2_O_3_, (x = 0, 1, 3 and 5 mol%) were prepared by the melt-quenching method using reagent-grade ZnO (Merck KGaA, Amsterdam, The Netherlands), WO_3_ (Merck KGaA, Darmstadt, Germany), H_3_BO_3_ (SIGMA-ALDRICH, St. Louis, MO, USA) and Eu_2_O_3_ (SIGMA-ALDRICH, St. Louis, MO, USA) as starting compounds. The homogenized batches were melted at 1240 °C for 30 min in a platinum crucible in air. The melts were cast into pre-heated graphite molds to obtain bulk samples. Then, the glasses were transferred to a laboratory electric furnace, annealed at 540 °C (a temperature 10 °C below the glass transition temperature) and cooled down to room temperature at a very slow cooling rate of about 0.5 °C/min in order to remove the thermal stresses. The amorphous state of the samples was confirmed by X-ray diffraction analysis (XRD) with a Bruker D8 Advance diffractometer, Karlsruhe, Germany, using Cu Kα radiation in the 10 < 2θ < 60 range. The glass transition temperature (T_g_) of the synthesized glasses was determined by differential thermal analysis (DTA) using a Setaram Labsys Evo 1600 apparatus (Setaram, Caluire-et-Cuire, France) at a heating rate of 10 K/min in air atmosphere. The density of the obtained glasses at room temperature was estimated by Archimedes’ principle using toluene (*ρ* = 0.867 g/cm^3^) as an immersion liquid on a Mettler Toledo electronic balance with sensitivity of 10^−4^ g. From the experimentally evaluated density values, the molar volume (*V_m_*), the molar volume of oxygen (*V_o_*) (volume of glass in which 1 mol of oxygen is contained) and the oxygen packing density (OPD) of glasses obtained were estimated using the following relations, respectively:(1)$$V_{m}=\frac{\sum x_{i}M_{i}}{\rho_{g}}$$
(2)$$V_{o}=V_{m}\times \left(\frac{1}{\sum x_{i}n_{i}}\right)$$
(3)$$OPD=1000\times C\times \left(\frac{\rho_{g}}{M}\right)$$
where *x_i_* is the molar fraction of each component *i*, *M_i_* the molecular weight, *ρ_g_* is the glass density, *n_i_* is the number of oxygen atoms in each oxide, *C* is the number of oxygens per formula units, and *M* is the total molecular weight of the glass compositions. The EPR analyses were carried out in the temperature range 120–295 K in X band at frequency 9.4 GHz on a spectrometer (Bruker EMX Premium, Karlsruhe, Germany). Optical transmission spectra at room temperature for the glasses were measured by spectrometer (Ocean optics, HR 4000, Duiven, The Netherlands) using a UV LED light source at 385 nm. Photoluminescence (PL) excitation and emission spectra at room temperature for all glasses were measured with a Spectrofluorometer FluoroLog3-22 (Horiba JobinYvon, Longjumeau, France). The IR spectra of the obtained samples were measured using the KBr pellet technique on a Nicolet-320 FTIR spectrometer (Madison, WI, USA) with a resolution of ±4 cm^−1^, by collecting 64 scans in the range 1600–400 cm^−1^. A random error in the center of the IR bands was found as ±3 cm^−1^. Raman spectra were recorded with a Raman spectrometer (Delta NU, Advantage NIR 785 nm, Midland, ON, Canada).

## 3. Results

### 3.1. XRD Spectra and Thermal Analysis

The amorphous nature of the prepared materials was confirmed by X-ray diffraction analysis. The measured X-ray diffraction patterns are shown in Figure 1. The photographic images (insets, Figure 1) show that transparent bulk glass specimens were obtained. The Eu^3+^-doped Nb_2_O_5_-free base zinc borate glass was colorless, while the glass samples having Nb_2_O_5_ were light yellowish due to the presence of Nb^5+^ ions [13].

The DTA data of investigated glasses are presented on Figure 2. All curves contain exothermic peaks over 500 °C corresponding to the glass transition temperature, T_g._ In the DTA lines, there is an absence of glass crystallization effects. However, the T_g_ values of Nb_2_O_5_-containing glasses were slightly lower as compared with the Eu^3+^-doped Nb_2_O_5_-free base zinc borate glass due to the formation of weaker Nb-O bonds (bond dissociation energy—753 kJ/mol) at the expense of stronger B-O bonds (bond dissociation energy—806 kJ/mol) [14].

### 3.2. Raman Analysis

The effect of Nb_2_O_5_ addition on the structure of glass 50ZnO:50B_2_O_3_:0.5Eu_2_O_3_ was studied by applying IR and Raman spectroscopy techniques. The Raman spectra of the 50ZnO:(50 − x)B_2_O_3_:xNb_2_O_5_:0.5Eu_2_O_3_, (x = 0, 1, 3 and 5 mol%) glasses are shown in Figure 3. 

The spectrum of Nb_2_O_5_-free glass (Figure 3, spectrum x = 0) agreed well with what has been reported by other authors for similar compositions [15,16,17]. The most prominent band at 877 cm^−1^ in the base binary glass x = 0 was assigned to the symmetric stretching of pyroborate dimers, [B_2_O_5_]^4−^ [15,16,17]. The two shoulders observed at 800 cm^−1^ and 770 cm^−1^ are due to the ring breathing of the boroxol rings and of the six-membered borate rings with one BO_4_ tetrahedron (tri-, tetra- and pentaborate rings), respectively [15]. The broad shoulder at about 705 cm^−1^ contains contributions of at least four borate arrangements: metaborate chains [BØ_2_O^−^]_n_ (deformation modes; Ø = bridging oxygen, O^−^ = nonbridging oxygen), in-plane and out-of-plane bending modes of both polymerized (BØ^0^) species and isolated orthoborate units (BO_3_)^3−^, and bending of the B-O-B connection in the pyroborate dimers, [B_2_O_5_]^4−^ [15,16,17]. The weak lower-frequency features at 270, 300 and 430 cm^−1^ are related to the Zn-O vibrations, Eu-O vibrations and borate network deformation modes, respectively [15,18]. The higher-frequency activity at 1235 cm^−1^ reflects the stretching of boron-non-bridging oxygen bonds, ν(B-O^−^) of the pyroborate dimers, while the other two features at 1365 and 1420 cm^−1^ are due to the B-O^−^ stretching in metaborate triangular units BØ_2_O^−^ [15]. The addition of Nb_2_O_5_ to the 50ZnO:50B_2_O_3_:0.5Eu_2_O_3_ glass led to the increase in the intensity of the bands at 705, 800 and 877 cm^−1^. Moreover, the shoulder at 800 cm^−1^ observed in the x = 0 glass spectrum became a peak in the Raman spectrum of glass having 1 mol% Nb_2_O_5_ (Figure 3 spectrum x = 1). With future increase in Nb_2_O_5_ content (Figure 3 spectrum x = 3 and x = 5), the peak at 800 cm^−1^ again turns into a shoulder. According to the Raman spectral data for the other niobium-containing glasses and crystalline compounds, the niobium can be present in the amorphous networks and in the crystalline structures in the form of NbO_4_ tetrahedral and octahedral NbO_6_ units with different degrees of polyhedral distortion and different kinds of connection (by corners or edges) [10,19]. Slightly and highly distorted octahedral units give rise to intensive bands in the regions 500–700 cm^−1^ and 850–1000 cm^−1^, respectively [10,19,20]. The vibration frequencies of NbO_4_ tetrahedra, that have been observed only in a few niobate crystals (LnNbO_4_, Ln = Y, Yb, La, Sm) and their melts containing NbO_4_ ions, occurred in the range 790–830 cm^−1^ [10,19,20,21]. In the 800–850 cm^−1^ range, stretching vibrations of Nb-O-Nb bonding in chains of corner-shared NbO_6_ are also reported [10,22]. On this basis, the increased intensity of the bands in the intermediate spectral range 600–1000 cm^−1^ observed in the spectra of Nb_2_O_5_-containing glasses compared to the Nb_2_O_5_-free glass is because of the overlapping contribution of the vibrational modes of niobate and borate structural groups present in the glass networks. The band at 800 cm^−1^ observed in the x = 1 glass is due to the coupled mode including the ring breathing of the boroxol rings, the symmetric stretching ν_1_ mode of tetrahedral NbO_4_ groups, and vibrations of Nb-O-Nb bonding [10,19]. Because of the complex character of this band, its transformation into a shoulder in the spectra of glasses x = 3 and x = 5 having higher Nb_2_O_5_ content is difficult to explain. However, the slight increase in the intensity of the low-frequency band at 430 cm^−1^ due to the bending (δ) vibrations of the NbO_6_ octahedra shows that with the increasing Nb_2_O_5_ concentration, NbO_4_ → NbO_6_ transformation takes place [23]. In addition, the reduced intensity of the band at 800 cm^−1^ observed in the glasses x = 3 and x = 5 also suggests decreasing numbers of NbO_4_ tetrahedra. This assumption is confirmed also by the variations in the physical parameters established, which will be discussed in the next paragraph of the paper. Stretching vibration ν_1_ of terminal Nb-O (short or non-bridging) bonds from NbO_6_ octahedra or short Nb-O bonds forming part of Nb-O-B bridges contribute to the band at 877 cm^−1^ [11]. The broad Raman shoulder at 705 cm^−1^ is attributed to the vibration of less-distorted NbO_6_ octahedra with no non-bridging oxygens, which overlap with the out-of-plane bending of triangular borate groups [10,15,16,17,19,24]. The nature of borate units also changes with the addition of Nb_2_O_5_ into the base x = 0 glass, which is manifested by the disappearance of the shoulder at 770 cm^−1^ due to the ring breathing of the six-membered borate rings with one BO_4_ tetrahedron (tri-, tetra- and pentaborate rings) together with the increased intensity of the band over 1200 cm^−1^ due to the vibration of trigonal borate units containing non-bridging oxygens. These spectral changes suggest that niobium oxygen polyhedra enter into the base zinc borate glass network by destruction of the superstructural borate units and favor formation of pyroborate [B_2_O_5_]^4−^ (band at 1235 cm^−1^) and metaborate BØ_2_O^−^ groups (bands at 1365 and at 1420 cm^−1^), which are charge-balanced by niobium.

### 3.3. IR Analysis

Information for the structure of the present glasses was also obtained by using IR spectroscopy. The normalized IR spectra of the glasses 50ZnO:(50 − x)B_2_O_3_:xNb_2_O_5_:0.5Eu_2_O_3_, (x = 0, 1, 3 and 5 mol%) are depicted in Figure 4. All glass spectra are characterized by a stronger absorption in the 1600–1150 cm^−1^ range, a wide spectral contour in the region 1150–750 cm^−1^ and strong bands in the 750–500 cm^−1^ range. IR spectra of Nb_2_O_5_-containing glasses (Figure 4, x = 1, x = 3, x = 5) exhibit also a band at 470 cm^−1^, reaching the highest intensity in the x = 3 glass spectrum. The stronger absorption in the 1600–1150 cm^−1^ range is connected with the stretching vibration of the B-O bonds in the trigonal borate units [25]. The IR activity in the spectral range 1150–750 cm^−1^ arises from the vibrations of B-O bonds in [BØ_4_]^−^ species, the vibrations of Nb-O-Nb bonding in chains of corner-shared NbO_6_ groups, and Nb-O short bond vibrations in highly distorted NbO_6_ octahedra and NbO_4_ tetrahedra [10,15,23,26]. The strong bands in the 750–500 cm^−1^ range are connected with the bending modes of trigonal borate entities that overlap with the ν_3_ asymmetric stretching vibrations of corner-shared NbO_6_ groups [10,23,26]. The low-frequency band at 470 cm^−1^, visible in the spectra of glasses containing Nb_2_O_5_ (x = 1, x = 3 and x = 5), can be related to the NbO_6_ stretching modes, having in mind the data in ref. [23] for Eu^3+^-doped crystalline rare-earth niobate Gd_3_NbO_7_. The structure of this compound consists of GdO_8_ units forming infinite chains along the [001] direction alternately with the NbO_6_ units and its IR spectrum containing the strong band at 483 cm^−1^ due to the stretching (ν) vibrations of NbO_6_ octahedra [23].

Analysis of the IR spectra obtained shows that various borate and niobate structural units co-exist in the structure of the investigated glasses and their vibrational modes are strongly overlapped. That is why a deconvolution process of the IR glass spectra was performed to make a more precise assignment of the peaks observed; the resulting spectra are shown in Figure 5.

The observed absorption bands in the deconvoluted spectra of the investigated glasses can be interpreted having in mind the band assignments proposed by Topper et al. in ref. [15] for xZnO-(1 − x)B_2_O_3_ glasses just above the metaborate stoichiometry, as well as taking into account our previous spectral investigation on 50ZnO:40B_2_O_3_:10WO_3_:xEu_2_O_3_ (0 ≤ x ≤ 10) and (50 − x)WO_3_:25La_2_O_5_:25B_2_O_3_:xNb_2_O_5_ (0 ≤ x ≤ 20) glasses reported in refs. [10,18]. Some other spectral data available in the literature for the similar glass and crystalline compounds were also taken into account [23,25,26,27]. The results are summarized in Table 1.

The IR data show that the addition of Nb_2_O_5_ into the 50ZnO:50B_2_O_3_ glass doped with 0.5 mol% (1.32 × 10^22^) Eu_2_O_3_ produces some changes in the IR spectrum, reflecting structural changes taking place with the composition. The most obvious effects are the reduction in the number of [BØ_4_]^−^ bands in the region 750–1150 cm^−1^, together with the strong decrease in the relative area of the band number 8 at 1236 cm^−1^ (stretching vibration of BØ_3_ triangles involved in various ring type superstructural borate groups). At the same time, new bands 13; 14; 15; 16 and 19 related to the vibrations of niobate structural units NbO_6_ and NbO_4_ (see Table 1) and 17, 18 connected with the vibration of pyro- and orthoborate groups in the network of Nb_2_O_5_-containing glasses appeared. The decreased number of bands due to the [BØ_4_]^−^ tetrahedra, and the strong reduction in band 8 at 1236 cm^−1^ (B-O-B bridges connecting superstructural groups through three-fold coordinated boron centers) are in agreement with the conclusions of the Raman analysis above and correspond to the destruction of borate superstructural units containing tetrahedral groups and increasing numbers of BO_3_-containing entities. On the other hand, the IR spectrum of Eu^3+^-doped crystalline Gd_3_NbO_7_ contains strong bands at 483 cm^−1^ and at 627 cm^−1^ (stretching vibration of NbO_6_) such as new bands 13 and 19 present in the IR spectra of Nb_2_O_5_-containing glasses. Since the spectral similarity supposes structural similarity, we suggest that the structure of investigated glasses is similar to the structure of the crystalline Gd_3_NbO_7_, which consists of infinite chains of GdO_8_ units alternately with the NbO_6_ units i.e., evidencing the presence of Eu^3+^ ions located around the niobate octahedra (Nb-O-Eu bonding) [23]. In the x = 3 glass spectrum, the band 13 at 480 cm^−1^ (ν of NbO_6_ in the vicinity of Eu^3+^) as well as the band 19 at 629 cm^−1^ possess higher relative area, indicating the highest number of NbO_6_ octahedra surrounding rare-earth ions in this glass composition (i.e., the highest number of Nb-O-Eu linkages).

Thus, the IR spectral analysis shows that addition of Nb_2_O_5_ into the base zinc borate glass depolymerizes the borate oxygen network, causing the destruction of superstructural borate groups and their conversion to BO_3_-containing borate entities. The structure of Nb_2_O_5_-containing glasses consists mainly of [BØ_2_O]^−^ and [BØ_4_]^−^ metaborate groups, [B_2_O_5_]^4−^ pyroborate and [BO_3_]^3−^ orthoborate units, isolated NbO_4_ tetrahedra and corner-shared NbO_6_. The presence of niobium increases the disorder and the degree of connectivity between the various structural units in the glass network, as it participates in the formation of mixed bridging Nb-O-B and Nb-O-Eu and as well as Nb-O-Nb linkages.

### 3.4. Physical Parameters

The observed variation in density and various physical parameters, such as molar volume (V_m_), oxygen molar volume (V_o_) and oxygen packing density (OPD), of the investigated glasses are listed in Table 2. They are in line with the proposed structural features, based on the Raman and IR spectral data. The Nb_2_O_5_-containing glasses are characterized by higher density and OPD values, evidencing that the presence of Nb_2_O_5_ in the zinc borate glass causes the formation of highly cross-linked and compact networks [28]. The lowest OPD value of the glass having the highest Nb_2_O_5_ content (x = 5), as compared with the OPD values of other Nb_2_O_5_-containing glasses, indicates decreasing cross-link efficiency of niobium ions and higher numbers of non-bridging atoms in the structure of this glass. With the introduction of 1 mol% Nb_2_O_5_ into the base zinc-borate glass, the molar volume V_m_ and oxygen molar volume V_o_ decrease, while with the further increase in Nb_2_O_5_ content (x = 3 and x = 5), both parameters start to increase. These observed changes can be explained with the NbO_4_ → NbO_6_ conversion upon Nb_2_O_5_ loading and the formation of a reticulated network because of the presence of high numbers of mixed bridging bonds (B-O-Nb, and Eu-O-Nb) within Nb_2_O_5_-containing glass networks [29].

### 3.5. Determination of Optical Band Gap

Some structural information also can be obtained from the optical band gap values (E_g_) evaluated from the UV-Vis spectra with the Tauc method by plotting (F(R_∞_) hν)^1/*n*^, *n* = 2 versus hν (incident photon energy), as shown in Figure 6 [30]. It is accepted that in metal oxides, the creation of non-bonding orbitals with higher energy than bonding ones shifts the valence band to higher energy, which results in E_g_ decreasing [31]. Therefore, the increase in the concentration of the NBOs (non-bridging oxygen ions) reduces the band gap energy. As seen from Figure 6, the E_g_ values decrease with increasing Nb_2_O_5_ content, indicating an increasing number of non-bridging oxygen species in the glass structure. This suggestion is in agreement also with the IR and Raman data obtained for the depolymerization of the borate network with the addition of Nb_2_O_5_ into the base ZnO-B_2_O_3_ glass. On the other hand, for the glasses containing Nb_2_O_5_, the reduction in E_g_ values is related to the increase in the glass’s overall polarizability due to the insertion of NbO_6_ octahedra and their mutual linking into the glass structure [8]. Thus, the same E_g_ values of x = 5 and x = 3 glasses show that there is an increasing number of polymerized NbO_6_ groups in the structure of glass x = 5.

### 3.6. EPR Spectroscopy

EPR analysis was carried out to provide insightful information about the Eu^2+^ ions in the studied glasses.

Figure 7 shows several dominant signals with g-values at g = 2.7, g = 4.6, g = 6.0. The most intensive feature is assigned to the impurities of isolated Mn^2+^ ions. The observed resonance signals in the spectral range 0–300 mT are assigned to the presence of Eu^2+^ ions in a highly asymmetric site environment [32,33]. The EPR spectra indicate the presence of low concentrations of Eu^2+^ ions in the obtained glasses, based on the comparison between the background spectrum and the analyzed spectra.

### 3.7. Optical Studies

The optical transmittance spectra and absorption coefficient data for investigated glasses are presented in Figure 8a,b.

As seen from Figure 8a, all glasses are characterized by good transmission in the visible region at around 80%. The low-intensity absorption bands at about 395 nm and 465 nm correspond to f-f transitions of Eu^3+^ ions between the ground and excited states. It should be mentioned that the reduction process of the valence of niobium ions (Nb^5+^ → Nb^4+^) produces very intense absorption peaks in the visible range due to the d-d transition. In the obtained spectra, there are no absorption bands corresponding to d-d transition, suggesting that Nb ions in the investigated glasses are present as Nb^5+^ only. The absorption coefficient (α) has been calculated with the following equation:$$\alpha =\left\{\ln\left(\frac{100}{T}\right)\right\}/d$$
where “*T*” is the percentage transmission and “*t*” is thickness of the glass. Figure 8b shows the absorption coefficients versus wavelength spectra. The maximum absorption values of the glasses increase with the increase in Nb_2_O_5_ content and vary between 290 and 316 nm.

### 3.8. Luminescent Properties

The excitation spectra (Figure 9) of the obtained glasses, monitored at 612 nm, consist of a wide excitation band below 350 nm and some narrow transitions of Eu^3+^ located at 317 nm (^7^F_0_ → ^5^H_3_), 360 nm (^7^F_0_ → ^5^D_4_), 375 nm (^7^F_0_ → ^5^G_2_), 380 nm (^7^F_1_ → ^5^L_7_), 392 nm (^7^F_0_ → ^5^L_6_), 413 nm (^7^F_0_ → ^5^D_3_), 463 nm (^7^F_0_ → ^5^D_2_) 524 (^7^F_0_ → ^5^D_1_), 530 nm (^7^F_1_ → ^5^D_1_) and 576nm (^7^F_0_ → ^5^D_0_) [34].The wide excitation band in the UV region is attributed to the charge transfer transition of Eu^3+^ (O^2−^ → Eu^3+^) [35,36,37,38] and host absorbing ZnO_n_ groups (O^2−^ → Zn^2+^) [39] and NbO_n_ groups (O^2−^ → Nb^5+^) [40]. Their contribution cannot be clearly differentiated due to the spectral overlap. 

Figure 9 shows that the increase in the Nb_2_O_5_ concentration in the glass composition leads to an increase in both charge transfer band intensity and narrow Eu^3+^ peaks. On the basis of structural analysis, it can be assumed that Nb_2_O_5_ modifies the glass network and makes it convenient for accommodation of Eu^3+^ ions. Hence, the incorporation of niobium into Eu^3+^-doped 50ZnO:50B_2_O_3_ host materials is favorable for achieving proper excitation, since, in general, Eu^3+^ bands are weak due to the parity-forbidden law. As can be seen from Figure 9, the strongest band is located at 392 nm (^7^F_0_ → ^5^L_6_ transition), followed by ^7^F_0_ → ^5^D_2_ transition at 463 nm. These data signify that the obtained phosphors can be efficiently excited with a range of excitation wavelengths of the commercially available near ultraviolet—NUV (250–400 nm) and blue LED chips (430–470 nm).

The emission spectra of Eu^3+^-doped 50ZnO:(50 − x)B_2_O_3_: xNb_2_O_5_:0.5Eu_2_O_3_:, x = 0, 1, 3 and 5 mol% glasses (Figure 10) were acquired upon excitation at 392 nm (^7^F_0_ → ^5^L_6_ transition). The observed bands are due to the intra-configurational transitions of the excited ^5^D_0_ state to the ground states ^7^F_0_ (578 nm), ^7^F_1_ (591 nm), ^7^F_2_ (612 nm), ^7^F_3_ (651 nm), and ^7^F_4_ (700 nm) in the ^4^F_6_ configuration of the Eu^3+^ ion [34]. The energy at 392 nm is not sufficient to excite the host optical groups, as their absorption is located below 350 nm, and thus, the non-radiative energy transfer to active ions cannot be expected. In detail, the excited ^5^L_6_ energy-level electrons relax into the first excited metastable singlet state ^5^D_0_ from ^5^D_3_, ^5^D_2_, and ^5^D_1_ states without visible emissions. In other words, the absorbed energy relaxes to the ^5^D_0_ state by the non-radiative process, and then, the emission of Eu^3+^ occurs by the radiative process.

The addition of Nb_2_O_5_ up to 3 mol% leads to an increase in the emission intensity. The luminescence suppression is observed at 5 mol% Nb_2_O_5_.

The strongest emission line, located at 612 nm, is caused by the forced electric dipole transition (ED) ^5^D_0_ → ^7^F_2_, sensitive to small changes in the environment, followed by the magnetic dipole (MD) ^5^D_0_ → ^7^F_1_ transition insensitive to the surroundings [34,35]. An indication that Eu^3+^ ions are distributed in a non-inversion symmetry sites in the glass host is the fact that the predominant emission is from the ED transition rather than from the MD transition. Therefore, the value of relative luminescent intensity ratio R of the two transitions (^5^D_0_ → ^7^F_2_)/(^5^D_0_ → ^7^F_1_) (Table 3) gives information on the degree of asymmetry around the Eu^3+^ ions [2,41]. The higher the value of the asymmetry parameter, the lower the local site symmetry of the active ion, and the higher Eu–O covalence and emission intensity. The calculated higher R values (from 4.31 to 5.16), compared to the others reported in the literature (Table 3) [18,42,43,44,45,46,47,48], suggest more asymmetry in the vicinity of Eu^3+^ ions, stronger Eu–O covalence, and thus enhanced emission intensity.

Comparing the R values of the synthesized zinc borate glass without Nb_2_O_5_ (4.31) and glass samples containing 1–5 mol% Nb_2_O_5_ (4.89–5.16), it can be assumed that Nb_2_O_5_ addition leads Eu^3+^ to a high-asymmetry environment in the host, increasing the intensity of ^5^D_0_ → ^7^F_2_ transition. The most intensive emission was registered with 3 mol% Nb_2_O_5_. Increasing the Nb_2_O_5_ content (5 mol%) leads to a slight decrease in the emission intensity (Figure 10) as a result of the increasing Eu^3+^ site symmetry (a slight reduction in R value) (Table 3). An additional indication of the Eu^3+^ location in non-centrosymmetric sites is the appearance of the ^5^D_0_ → ^7^F_0_ transition in the emission spectra. Based on the standard Judd-Ofelt theory, this transition is strictly forbidden. According to Binnemans, the observation of the ^5^D_0_ → ^7^F_0_ band shows that Eu^3+^ ions occupy sites with C_2v_, C_n_ or C_s_ symmetry [49].

#### CIE Color Coordinates and CCT (K) Values

To characterize the emission color of Eu^3+^-doped glasses, the standard Commission International de l’Eclairage (CIE) 1931 chromaticity diagram was applied [50]. From the luminescence spectra, the chromaticity coordinates of specimens were calculated using color calculator software SpectraChroma (CIE coordinate calculator) [51]. The obtained values are listed in Table 4, whereas references are included for the chromaticity coordinates of the commercial phosphor Y_2_O_2_S:Eu^3+^ [52] and National Television Standards Committee (NTSC) for red color. As can be seen from Table 4, the chromaticity coordinates of the niobium-containing glasses are very close to the standard recommended by NTSC (0.67, 0.33) values and nearly equivalent to the commercially applied red phosphor Y_2_O_2_S:Eu^3+^ (0.658, 0.340). The calculated values are almost identical and cannot be individually separated on the CIE diagram (Figure 11). These data show that the obtained glasses are characterized by high color purity.

The color-correlated temperature (CCT) was calculated by the McCamy empirical formula [53]:$$CCT=-449n^{3}+3525n^{2}-6823n+5520.33$$
where *n* = (*x* − *x_e_*)/(*y* − *y_e_*) is the reciprocal slope, (*x_e_* = 0.332, *y_e_* = 0.186) is the epicenter of convergence, and *x* and *y* are the chromaticity coordinates. The phosphors with CCT values below 3200 K are generally considered as a warm light source, while those with values above 4000 K, as a cold light source [53]. The calculated CCT values of Eu^3+^-doped glasses (Table 4) range from 2301.26 K to 2518.60 K, and these glasses can be considered as warm red light-emitting materials for solid-state lighting applications.

## 4. Discussion

The Raman and IR spectral data as well as the established values of the structurally sensitive physical parameters demonstrate that at smaller concentrations (up to 5 mol%), the niobium ions are embedded into the base Eu^3+^: ZnO:B_2_O_3_ glass as isolated NbO_4_ tetrahedra and corner-shared NbO_6_ with increasing distortion upon Nb_2_O_5_ loading. NbO_4_ tetrahedral units play a network-forming role and strengthened the host glass structure through B-O-Nb bonding. NbO_6_ octahedra are situated around the Eu^3+^ ions (i.e., niobate groups are charge-balanced by Eu^3+^ ions), and the higher numbers of NbO_6_ surrounding Eu^3+^ are found for the glass containing 3 mol% Nb_2_O_5_. Other than by Eu^3+^ ions, NbO_6_ octahedra are also charge-balanced by Zn^2+^ ions. Hence, the incorporation of Nb_2_O_5_ into Eu^3+^: ZnO:B_2_O_3_ glass creates more disordered and reticulated glass networks, which are favorable for doping with Eu^3+^ active ions. Moreover, the DTA analysis shows high values of glass transition temperatures (over 500 °C) and also an absence of glass crystallization effects—both confirming the formation of connected and stable glass networks. 

The observed optical properties are discussed on the basis of the glass structural features. The most intensive Eu^3+^ emission peak, corresponding to the hypersensitive ^5^D_0_ → ^7^F_2_ transition, along with the high values of the luminescent ratio R, evidence that Eu^3+^ ions are located in low site symmetry in the host matrix. This emission peak intensity and the R values of Nb_2_O_5_-containing glasses are higher in comparison with the Nb_2_O_5_-free Eu^3+^: ZnO:B_2_O_3_ glass, indicating that Eu^3+^ ions are in higher-asymmetry environments in the Nb_2_O_5_-containing glasses because of the combination of niobate and borate structural units in the active ion surroundings. Thus, the introduction of Nb_2_O_5_ oxide into the Eu^3+^: ZnO:B_2_O_3_ glass increases connectivity in the glass network and contributes to the creation of a more distorted and rigid glass structure that lowers the site symmetry of the rare-earth ion and improves its photoluminescence behavior. The influence of Eu^2+^ ions on the luminescence of Eu^3+^ is negligible due to their low content.

The results of these investigations show that Nb_2_O_5_ is an appropriate constituent for modification of zinc borate glass structure and for enhancing the luminescent intensity of the doped Eu^3+^ ion.

## 5. Conclusions

The impact of the glass matrix on the luminescent efficiency of europium has been studied. According to IR and Raman data, the structure of glasses consists of [BØ_2_O]^−^ and [BØ_4_]^−^ metaborate groups, [B_2_O_5_]^4−^ pyroborate and [BO_3_]^3−^ orthoborate units, isolated NbO_4_ tetrahedra and corner-shared NbO_6_. The local environment of the Eu^3+^ ions in the Nb_2_O_5_-containing ZnO:B_2_O_3_ glasses is dominated by the interaction with both, borate and NbO_6_ octahedral structural groups. The luminescent properties of the obtained Eu^3+^-doped glasses revealed that they could be excited by 392 nm and exhibit pure red emission centered at 612 nm (^5^D_0_ → ^7^F_2_ transition). The incorporation of niobium oxide into the ZnO:B_2_O_3_ glass enhances the luminescent intensity, making it a desirable component in the glass structure. It was established that the optimum Nb_2_O_5_ concentration to obtain the most intensive red luminescence is 3 mol%. The structure–optical property relationship studied in this work will be favorable for the elaboration of novel red-emitting materials.

## Figures and Tables

**Figure 1 materials-17-01415-f001:**
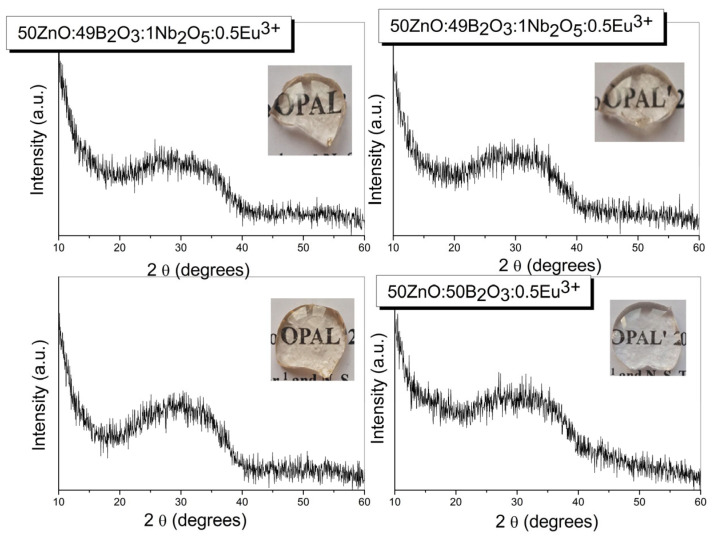
XRD patterns of glasses 50ZnO:(50 − x)B_2_O_3_:0.5Eu_2_O_3_:xNb_2_O_5_, (x = 0, 1, 3 and 5 mol%).

**Figure 2 materials-17-01415-f002:**
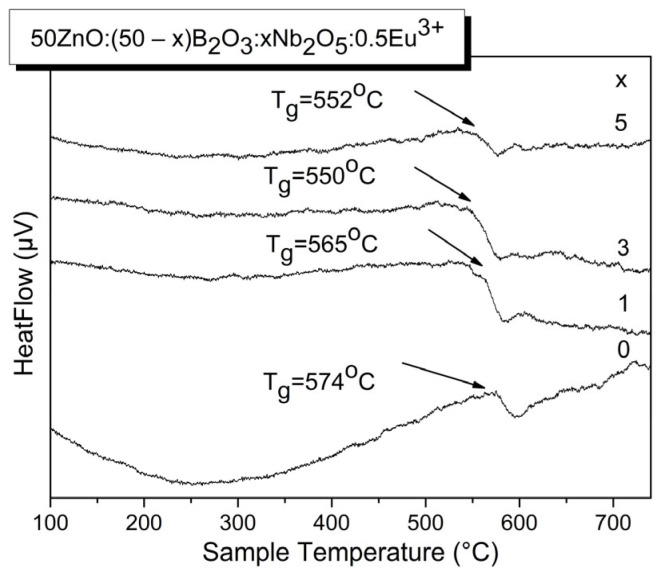
DTA curves of glasses 50ZnO:(50 − x)B_2_O_3_:0.5Eu_2_O_3_:xNb_2_O_5_, (x = 0, 1, 3 and 5 mol%).

**Figure 3 materials-17-01415-f003:**
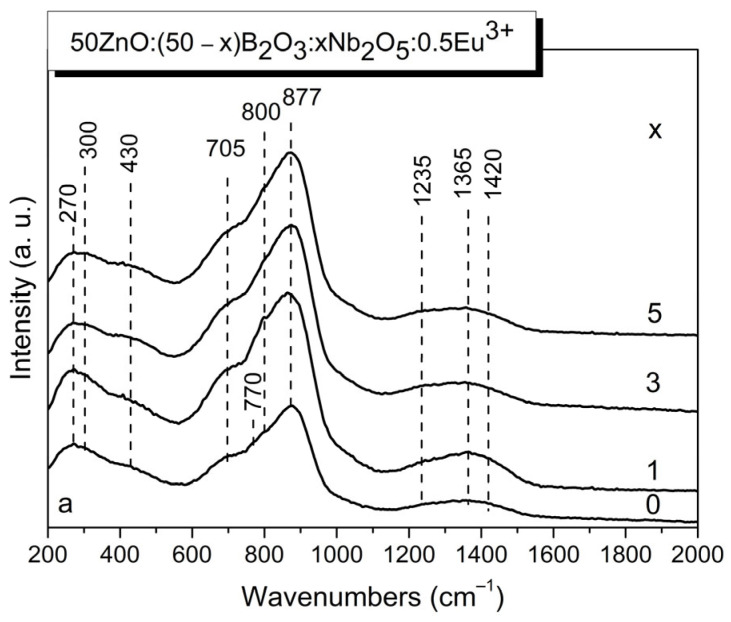
Raman spectra of glasses 50ZnO:(50 − x)B_2_O_3_:0.5Eu_2_O_3_:xNb_2_O_5_, (x = 0, 1, 3 and 5 mol%).

**Figure 4 materials-17-01415-f004:**
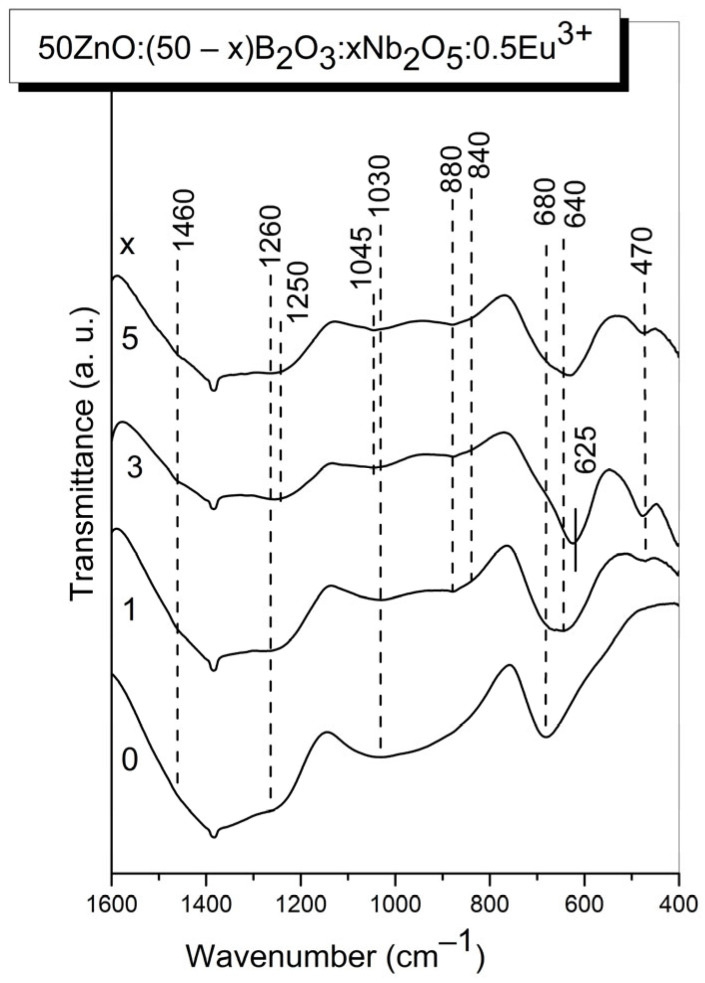
IR spectra of glasses 50ZnO:(50 − x)B_2_O_3_:0.5Eu_2_O_3_:xNb_2_O_5_, (x = 0, 1, 3 and 5 mol%).

**Figure 5 materials-17-01415-f005:**
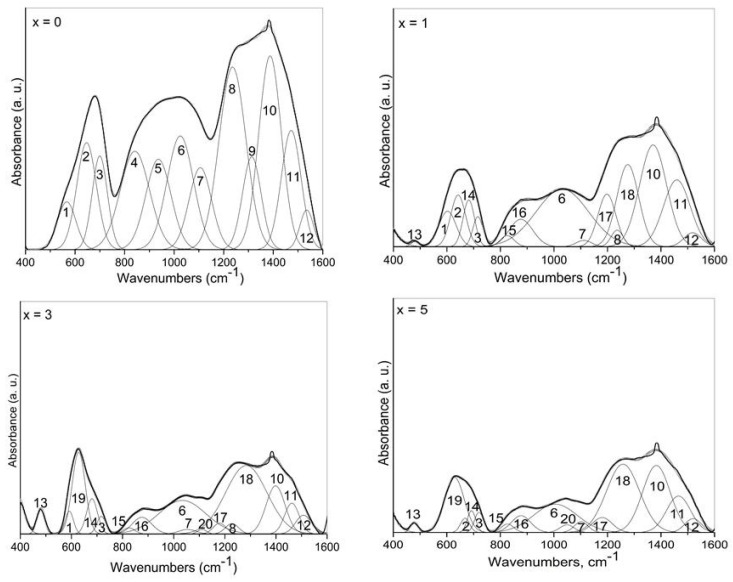
Deconvoluted IR spectra of glasses 50ZnO:(50 − x)B_2_O_3_:0.5Eu_2_O_3_:xNb_2_O_5_, (x = 0, 1, 3 and 5 mol%).

**Figure 6 materials-17-01415-f006:**
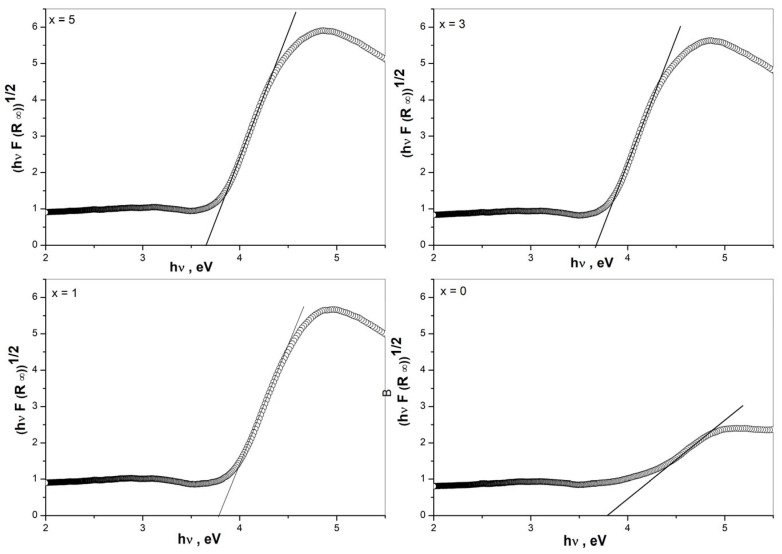
Tauc’s plots of glasses 50ZnO:(50 − x)B_2_O_3_:0.5Eu_2_O_3_:xNb_2_O_5_, (x = 0, 1, 3 and 5 mol%).

**Figure 7 materials-17-01415-f007:**
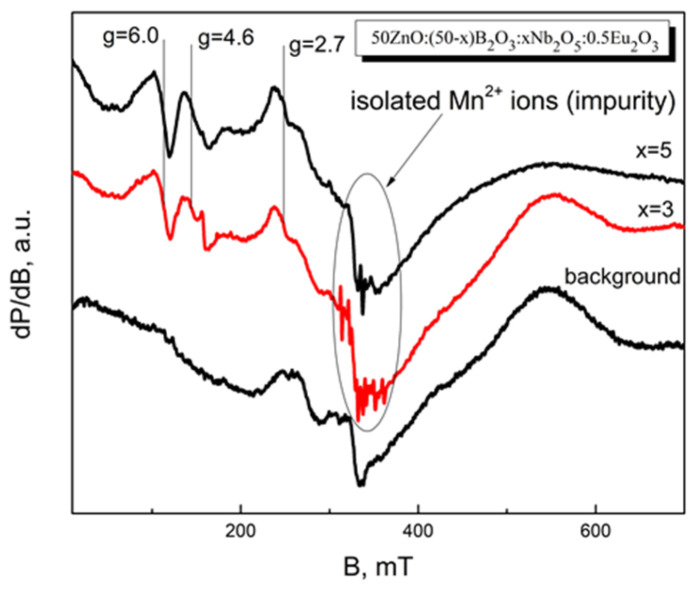
EPR spectra of glasses 50ZnO:(50 − x)B_2_O_3_:0.5Eu_2_O_3_:xNb_2_O_5_, (x = 3 and 5 mol%).

**Figure 8 materials-17-01415-f008:**
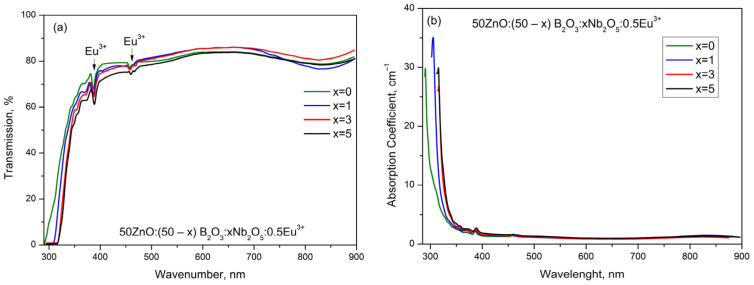
(**a**) Optical transmission spectra at room temperature; (**b**) absorption coefficient in the range of 250 nm–900 nm for glasses 50ZnO:(50 − x)B_2_O_3_:0.5Eu_2_O_3_:xNb_2_O_5_, (x = 0, 1, 3 and 5 mol%).

**Figure 9 materials-17-01415-f009:**
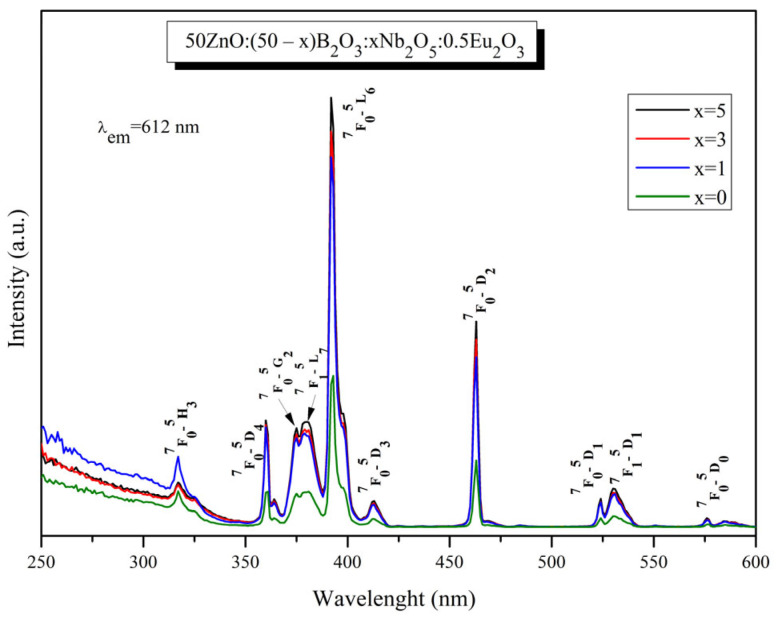
Excitation spectra of 50ZnO:(50 − x)B_2_O_3_:xNb_2_O_5_:0.5Eu_2_O_3_ (x = 0, 1, 3 and 5 mol%) glasses.

**Figure 10 materials-17-01415-f010:**
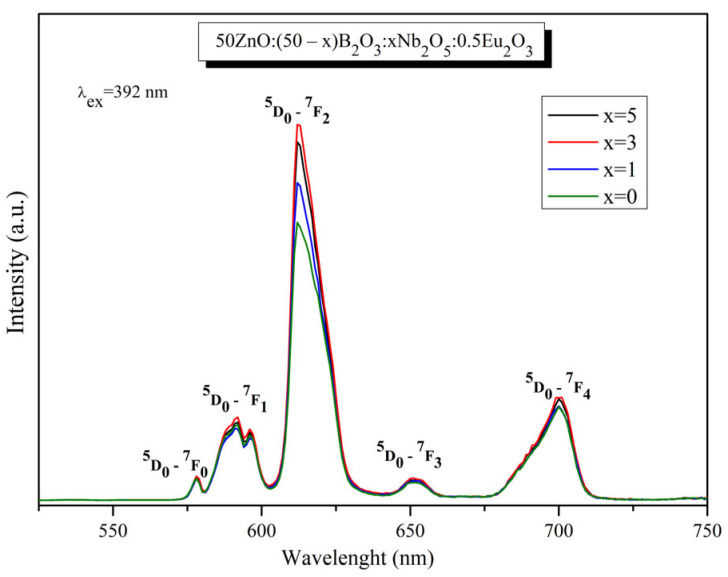
Emission spectra of 50ZnO:(50 − x)B_2_O_3_: xNb_2_O_5_:0.5Eu_2_O_3_:, x = 0, 1, 3 and 5 mol% glasses.

**Figure 11 materials-17-01415-f011:**
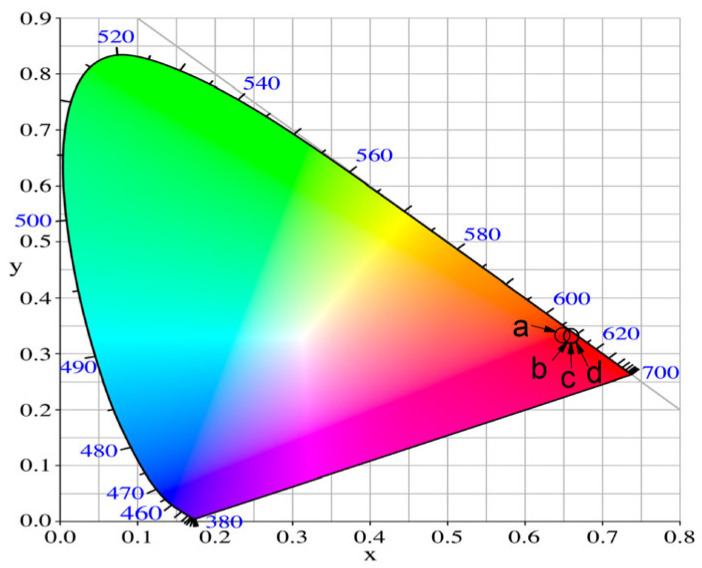
CIE chromaticity diagram of 50ZnO:(50 − x)B_2_O_3_: xNb_2_O_5_:0.5Eu_2_O_3_ (a) x = 0, (b) x = 1, (c) x = 3, (d) x = 5 glasses.

**Table 1 materials-17-01415-t001:** IR peak positions of the deconvoluted IR spectra of glasses 50ZnO:(50 − x)B_2_O_3_:0.5Eu_2_O_3_:xNb_2_O_5_, (x = 0, 1, 3 and 5 mol%) and their assignments.

Peak #	Peak Position, cm^−1^				Band Assignment	Ref.
	0% Nb_2_O_5_	1% Nb_2_O_5_	3% Nb_2_O_5_	5% Nb_2_O_5_		
1	566	602	591	-	Bending modes of various trigonal borate units.	[15,18]
2	646	641	-	668	Bending modes of various trigonal borate units.	[15,18]
3	700	715	717	720	Bending modes of various trigonal borate units.	[15,18]
4	841	-	-	-	B-O stretching modes of [BØ]^−^ from ring type superstructures containing one or two tetrahedral boron sites + ν_as_ of tetrahedral metaborate groups.	[15]
5	936	-	-	-	B-O stretching modes of [BØ]^−^ from ring type superstructures containing one or two tetrahedral boron sites + ν_as_ of tetrahedral metaborate groups.	[15]
6	1025	1031	1034	1005	B-O stretching modes of [BØ]^−^ from ring type superstructures containing one or two tetrahedral boron sites + ν_as_ of tetrahedral metaborate groups.	[15]
7	1105	1111	1111	1101	B-O stretching modes of [BØ]^−^ from ring type superstructures containing one or two tetrahedral boron sites + ν_as_ of tetrahedral metaborate groups.	[15]
8	1236	1236	1228	-	Stretching vibrations of BØ_3_ triangles involved in various ring type superstructural borate groups (boroxol rings, tri-,tetra- and pentaborates).	[15,18]
9	1313	-	-	-	B-O^−^ stretch in pyroborate units.	[25]
10	1387	1371	1398	1383	Stretching vibrations of non-bridging B-O^−^ bonds in metaborate units, BØ_2_O^−^.	[10,15]
11	1472	1459	1462	1466	Stretching vibrations of non-bridging B-O^−^ bonds in metaborate units, BØ_2_O^−^.	[10,15]
12	1535	1519	1508	1520	Stretching of B-Ø bonds in neutral BØ_3_ triangles.	[25]
13	-	480	481	479	Stretching vibrations of NbO_6_.	[23,27]
14	-	681	680	692	ν_3_ asymmetric stretching vibrations of NbO_6_.	[10]
15	-	825	825	827	Vibrations of Nb-O-Nb bonding in chains of corner shared NbO_6_ groups.	[10]
16	-	876	876	877	ν_1_ symmetric mode of short Nb-O bonds in distorted NbO_6_ and NbO_4_ units.	[10]
17	-	1197	1179	1180	B-O-B stretch in pyroborate units, BØO_2_^2−^.	[25]
18	-	1276	1284	1258	ν_3_ asymmetric stretching mode of orthoborate groups, BO_3_^3−^.	[25]
19	-	-	628	629	ν_3_ asymmetric stretching vibrations of NbO_6_.	[10,26]
20	-	1054	1054	1049	B-O stretching modes of [BØ]^−^ from ring type superstructures containing one or two tetrahedral boron sites + ν_as_ of tetrahedral metaborate groups.	[15]

**Table 2 materials-17-01415-t002:** Values of physical parameters of glasses 50ZnO:(50 − x)B_2_O_3_:0.5Eu_2_O_3_:xNb_2_O_5_, (x = 0, 1, 3 and 5 mol%): density (ρ_g_), molar volume (V_m_), oxygen molar volume (V_o_), oxygen packing density (OPD). Optical band gap (E_g_) values of glasses 50ZnO:(50 − x)B_2_O_3_:0.5Eu_2_O_3_:xNb_2_O_5_, (x = 0, 1, 3 and 5 mol%).

Sample ID	ρ_g_ (g/cm^3^)	V_m_ (cm^3^/mol)	V_o_ (cm^3^/mol)	OPD (g atom/L)	E_g_ (eV)
x = 0	3.413 ± 0.001	22.634	11.261	88.804	3.80
x = 1	3.567 ± 0.001	22.208	10.940	91.408	3.78
x = 3	3.663 ± 0.001	22.697	10.965	91.201	3.67
x = 5	3.665 ± 0.001	23.755	11.258	88.823	3.66

**Table 3 materials-17-01415-t003:** Relative luminescent intensity ratio (R) of the two transitions (^5^D_0_ → ^7^F_2_)/(^5^D_0_ → ^7^F_1_) for glasses with different Nb_2_O_5_ content and of other reported Eu^3+^-doped oxide glasses.

Glass Composition	Relative Luminescent Intensity Ratio, R	Reference
50ZnO:50B_2_O_3_:0.5Eu_2_O_3_	4.31	Present work
50ZnO:49B_2_O_3_:1Nb_2_O_3_:0.5Eu_2_O_3_	4.89	Present work
50ZnO:47B_2_O_3_:3Nb_2_O_3_:0.5Eu_2_O_3_	5.16	Present work
50ZnO:45B_2_O_3_:5Nb_2_O_3_:0.5Eu_2_O_3_	5.11	Present work
50ZnO:40B_2_O_3_:10WO3:xEu_2_O_3_ (0 ≤ x≤ 10)	4.54–5.77	[18]
50ZnO:40B_2_O_3_:5WO_3_:5Nb_2_O_5_:xEu_2_O_3_ (0 ≤ x ≤ 10)	5.09–5.76	[42]
(100 − y)TeO_2_-10Nb_2_O_5_-yPbF_2_ (0 ≤ y ≤ 30)	2–4.16	[43]
69TeO_2_:1K_2_O:15Nb_2_O_5_:1.0Eu_2_O_3_	5	[44]
60TeO_2_:19ZnO:7.5Na_2_O:7.5Li_2_O:5Nb_2_O_5_:1Eu_2_O_3_	3.73	[45]
4ZnO:3B_2_O_3_:0.5–2.5 mol% Eu^3+^	3.94–2.74	[46]
(99.5 − x):B_2_O_3_:xLi_2_O:0.5Eu_2_O_3_	2.41–3.40	[47]
(64 − x)GeO_2_:xSiO_2_:16K_2_O:6BaO:4Eu_2_O_3_	3.42–4.07	[47]
(98 − x)P_2_O_5_:xCaO:2Eu_2_O_3_	3.88–3.95	[47]
79TeO_2_ + 20Li_2_CO_3_ + 1Eu_2_O_3_	4.28	[48]

**Table 4 materials-17-01415-t004:** CIE chromaticity coordinates, dominant wavelength, color purities and correlated color temperature (CCT, K) of 50ZnO:(50 − x)B_2_O_3_: xNb_2_O_5_:0.5Eu_2_O_3_, x = 0, 1, 3 and 5 mol%.

Glass Composition	Chromaticity Coordinates (x, y)	CCT (K)
50ZnO:B_2_O_3_:0.5Eu_2_O_3_ (x = 0)	(0.645, 0.346)	2301.26
50ZnO:49B_2_O_3_:1Nb_2_O_3_:0.5Eu_2_O_3_ (x = 1)	(0.656, 0.344)	2479.99
50ZnO:47B_2_O_3_:3Nb_2_O_3_:0.5Eu_2_O_3_ (x = 3)	(0.656, 0.343)	2505.78
50ZnO:45B_2_O_3_: 5Nb_2_O_3_:0.5Eu_2_O_3_ (x = 5)	(0.657, 0.343)	2518.60
NTSC standard for red phosphors	(0.67, 0.33)	
Y_2_O_2_S:Eu^3+^	(0.658, 0.340)	

## Data Availability

Data are contained within the article.

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
