# Peer review of "Structure and Luminescent Properties of Niobium-Modified ZnO-B_2_O_3_:Eu^3+^ Glass"

_materials, 2024, doi:10.3390/ma17061415_

Round 1

Reviewer 1 Report

Comments and Suggestions for Authors

The submitted paper deals with structural and spectroscopic properties of binary glass system modified by niobium oxide and doped with europium luminescent ions. This paper reveals some meaningful result especially related to structural glass features. Some comments are addressed to the Authors:

-On Figure 8 aperture noises are higher than Eu3+ optical transitions, hence it is unacceptable and the quality of these spectra should be corrected. Furthermore, Figure 8b presenting absorption coefficient of the investigated glasses can be added as well.

-Energy transfer processes are discussed. Accordingly, the comments attributed to relaxation dynamic of the involved Eu excited states and host-created centers are needed.

-The non-radiative energy transfer from niobate groups to the Eu3+ ions is rather doubtful and more detailed study including the contribution of other competitive optical centers (Eu2+, O-Eu charge transfer transitions) and relaxation dynamic of the involved excited states should be prepared.

-Page 10, “Under excitation at 612 nm, the appearance of absorption host was observed”, rather “The absorption of host contributes to 612 nm Eu3+ luminescence excitation”. Please correct it.

-The values of estimated fundamental energy gap for materials under study can be incorporated to Table 1.

-The reference explaining “weaker Nb-O bond at the expense of stronger B-O should be indicated.

-The quality of Figure 5 is low and can be improved.

-Concentration of europium ions may be expressed in ions/cm3.

-Abstract – “sharp” 7F0-5L0 transition of Eu3+ is unfortune statement.

Reviewer 2 Report

Comments and Suggestions for Authors

The manuscript entitled “Structure and luminescent properties of niobium modified ZnO-B2O3:Eu3+ glass” deals with effect of niobium on structure and optical properties of the zinc-borate glass. The presented results are interesting but discussion has some issues those should be clarified before publication. So, I recommend a Major Revision. Here some comments.

 1.              The most questionable suggestion of the authors is “NbO6 octahedra in the vicinity of Eu3+ ions in its glass network”. It is not obvious that at even 5 mol % of Nb2O5 and 0.5 mol % Eu2O3 the probability of Eu to be near Nb is quite high. It is not supported by data as for me.

2.              Figure 5 has low resolution. It is looks like that bottom parts are the same (missed x=3). Why the spectra take only half of the space at some parts of the figure 5 while absorbance in a.u.?

3.              A discussion about fig.5 is very extensive. It would be clearer for reader when results of the spectra decomposition are collected into table like:

Peak #

Peak position, cm-1

Band assignment

Reference

0% Nb2O5

1% Nb2O5

3% Nb2O5

5% Nb2O5

with smaller discussion on changes with increasing of Nb2O5 content

 4.              Line 296. Authors said “Some structural information can be also obtained from the optical band gap values”. What structural information have been obtained from this data?

5.              Figure 6. Vertical axis at plot for x=0 starts not from 0. Maybe Eg for x=0 also not correct.

6.              Lines 334-345. “The wide excitation band in the UV-region is attributed to the charge transfer transition of Eu3+ (O2-> Eu3+) for x=0”. – It is more likely that charge transfer band overlaps with glass host absorption.

7.              Lines 345-352. Authors compare glass under study (low content of Nb) with literature data on niobates – it is not correct. Why only NbOn groups ascribed to host absorbing groups? How about BOx and ZnOx? Absorption at wavelength < 350 nm (fig. 8) are quite similar for all the samples, including those without niobium. “Charge transfer band” is superimposed light absorption by different components. More likely, that Nb2O5 modify this band through modification of glass network than through energy transfer from NbOn to Eu3+.

8.              Lines 366-372. Absence of NbOn groups when excitation takes place at 392 is expected as in cited references the emission in niobates effectively excited at 300 nm.

9.              Lines 451-453. It is said “Finally, the existence of NbO6 groups around Eu3+ ions ensures an occurrence of non-radiative energy transfer from the niobate groups to the active ions that additionally improves the Eu3+luminescence intensity.” Any calculation of average distances between Nb and Eu3+ ions?

10.           Authors used composition 50ZnO:(50-x)B2O3:0.5Eu2O3:xNb2O5, (x=0, 1, 3 and 5 mol %) that gives 100.5 mol %. Please clarify.

11.           Line 190. “upon Nb2O5 content” probably means “upon increasing of Nb2O5 content”

12.           Check the typos, like “trtrahedra” in Abstract and (16-19) instead of [16-19] at line 110.

Round 2

Reviewer 1 Report

Comments and Suggestions for Authors

This paper can be accepted in its present form.

Reviewer 2 Report

Comments and Suggestions for Authors

The revised manuscript can be accepted for publication as is.